# 9-Cyanopyronin probe palette for super-multiplexed vibrational imaging

Yupeng Miao [1], Naixin Qian[1], Lixue Shi [1], Fanghao Hu[1] & Wei Min [1,2 ✉]

Multiplexed optical imaging provides holistic visualization on a vast number of molecular targets, which has become increasingly essential for understanding complex biological processes and interactions. Vibrational microscopy has great potential owing to the sharp linewidth of vibrational spectra. In 2017, we demonstrated the coupling between electronic pre-resonant stimulated Raman scattering (epr-SRS) microscopy with a proposed library of 9-cyanopyronin-based dyes, named Manhattan Raman Scattering (MARS). Herein, we develop robust synthetic methodology to build MARS probes with different core atoms, expansion ring numbers, and stable isotope substitutions. We discover a predictive model to correlate their vibrational frequencies with structures, which guides rational design of MARS dyes with desirable Raman shifts. An expanded library of MARS probes with diverse functionalities is constructed. When coupled with epr-SRS microscopy, these MARS probes allow us to demonstrate not only many versatile labeling modalities but also increased multiplexing capacity. Hence, this work opens up next-generation vibrational imaging with greater utilities.

---

[1] Department of Chemistry, Columbia University, New York, NY, USA. [2] Kavli Institute for Brain Science, Columbia University, New York, NY, USA. ✉email: wm2256@columbia.edu

Fluorescence microscopy has served the biomedical imaging community for decades[1–3]. However, one of the bottlenecks with fluorescence-based techniques is the limited ability for multiplexing. The fast electronic dephasing of fluorophores at ambient temperature dramatically broadens their excitation and emission spectra to 50–100 nm, resulting in the accommodation of no more than 6 fluorophores simultaneously[4–6]. This so-called "color barrier" becomes increasingly problematic, as we enter the blooming era of system biology in which gathering multi-parameter information is critical in broad applications such as profiling brain circuits, building tissue atlases, and phenotyping tumor heterogeneity[7–10]. Although sequential labeling and imaging, such as cyclic immunofluorescence and DNA exchange-based multiplexing (Exchange-PAINT), was developed to expand the multiplexing capacity[11–14], these methods cannot work with live samples and they are clearly time- and labor-consuming. Moreover, these repeated cycles likely lead to accumulative structural changes that alter epitope antigenicity[15] and cause physical distortion or even histology damage that prevents accurate image co-registration across repeated staining cycles.

Vibrational microscopy provides a direct physical solution to break the "color barrier" with great promise, because vibrational signatures of molecules are inherently much narrower than fluorescence spectrum[16–19]. For example, Cy5 dye shows a $\sim 500\,cm^{-1}$ FWHM in the emission spectrum, but the Raman signature of its double bond stretching mode exhibits FWHM of merely $10\,cm^{-1}$[18]. Therefore, employing Raman signal as imaging contrast could potentially offer many more colors than fluorescence. Yet, the sensitivity of Raman microscopy (typically in the mM range) is far from ideal for imaging specific proteins inside cells. In 2017, we developed a nonlinear Raman imaging technique named electronic pre-resonance Stimulated Raman Scattering Microscopy (epr-SRS) (Fig. 1a)[18]. When the pump laser energy approaches fluorophores' electronic transition but remains detuned from the rigorous resonance, the vibrational signal from the fluorophore backbone can be enhanced by $>10^4$ through the vibrational and electronic coupling[20]. Harnessing this electronic pre-resonance effect with SRS microscopy leads to a desirable combination of high detection sensitivity (down to

250 nM) and fine Raman contrast ($\sim 10\,cm^{-1}$). To create the matching imaging probes with resolvable Raman peaks, a dye palette named Manhattan Raman scattering (MARS) was proposed, each containing a triple bond (such as nitriles and alkynes) in the conjugated system[18]. As such, epr-SRS microscopy of MARS probes has emerged as an ultrasensitive method to image specific biological targets inside cells with vibrational contrast.

However, as the key component of the super-multiplexed imaging technology, MARS probes palette is in its infancy for several reasons that are intertwined. First, regarding synthetic chemistry, we lacked robust and efficient methods to synthesize MARS dyes from common starting materials. In our prior report, most of the MARS dyes had to rely on commercial pyronin dyes which are generally short of choices, and the synthesis route was poor in atom economy (e.g., MARS2237 isotopologues in Supplementary Fig. 1). Second, regarding physical chemistry and rational design, the structure-spectroscopy relationship of MARS dyes is elusive. Such lack of knowledge prevented us from expanding the MARS palette through rational design. Third, regarding the chemical biology of probe development, both the number and the type of available functionalized probes were severely limited. Because of the reliance on commercial pyronin dyes, the proposed MARS dyes had to be symmetric in structure and consequentially had no functionalizable sidechains for targeting capability—in other words, they were just dyes but not imaging probes yet. In fact, there were only 4 NHS ester functionalized probes (all in the same type) in the original MARS palette. Other broadly used labeling techniques such as click chemistry were not explored, and the imaging application scope was severely restrained.

To address the underlying synthetic chemistry challenges, herein we have developed robust methods for synthesizing MARS dyes of different core atoms, conjugation ring numbers, and stable isotope substitutions with great efficiency. Systematic spectroscopy study on the newly synthesized dyes has revealed four rules governing the vibrational tuning mechanisms by diverse structure features. Remarkably, a quantitative model can be established by integrating these four rules to predict vibrational frequencies directly from overall MARS dye structures. We

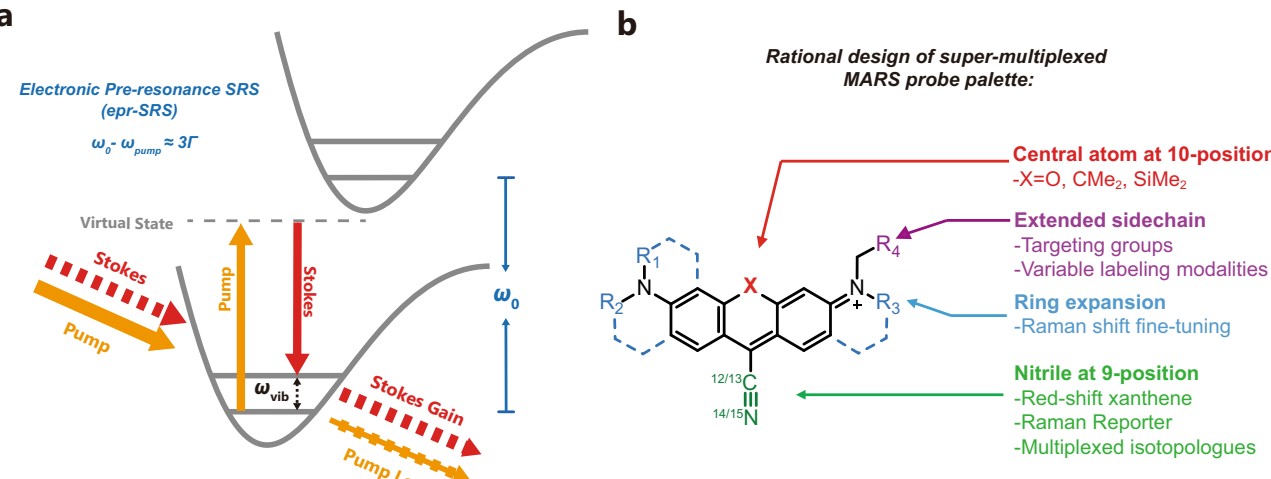

**Fig. 1 Spectroscopy principle and probe design of electronic pre-resonance SRS microscopy. a** Schematic illustration of electronic pre-resonance SRS. Two synchronized laser beams were tightly focused on molecules of interest. When the energy difference between two photons matches vibrational transition, one pump photon can be converted to stokes photon and the intensity fluctuation will be detected. As pump energy approaches electronic transition (pre-resonance), such process will be greatly enhanced. $\omega_0$, $\omega_{pump}$, and $\omega_{vib}$ represents for frequencies of electronic transition, pump beam photon, and vibrational transition, respectively. $\Gamma$ is the homogeneous linewidth, typically around $700\,cm^{-1}$. **b** Design and engineering principles of 9-cyanopyronin library. Three key structural features were rationally tuned to generate new MARS dye library. One sidechain was installed on the amino group to facilitate facile functionalization.

then employed this structure-spectroscopy model to rationally design MARS probes that are spectrally resolvable with existing probes. Moreover, the robust synthetic methods allowed us to readily derivatize asymmetric dye structure and introduce a set of 30 MARS probes (more than 10 types) with specific labeling capability. With this greatly expanded MARS probe palette, we have demonstrated a variety of epr-SRS imaging applications in cells and tissues including many new versatile labeling modalities (such as click chemistry, peptide, and organelle labels) and increased multiplexing capacity than previously available. Therefore our work has established integrated framework for the synthetic chemistry, physical chemistry and chemical biology of multicolor vibrational probes, paving the way for next-generation super-multiplexed imaging.

## Results

**General design principles for functionalized MARS probes**. 9-cyano xanthene was first reported in 1993[21]. It's intriguing that the substitutions at C-9 position play important roles in the wavelengths of absorption maxima, especially in the cases of electron-withdrawing groups (EWGs)[22,23]. One of the most well-known fluorophores, rhodamine bears a phenyl group at C-9 position bringing a+ 20 nm bathochromic shift. As a comparison, the cyano group causes an even remarkable shift of more than 100 nm, pushing the dye into the near-infrared region (>650 nm). We realized these nitrile-bearing molecules can fulfill the requirements of both (1) long-wavelength absorption for necessary epr-SRS excitation and (2) triple bonds for bio-orthogonal Raman imaging in the desirable cell-silent region. Therefore, 9-cyano xanthene was selected as the parent structure to generate MARS dye library for epr-SRS microscopy.

To develop systematic synthetic methodology, we sought to modify 9-cyano xanthene dyes by changing four key structural features:[24] (Fig. 1b) (1) The mass of the cyano group by means of isotope doping on CN bond. As described in the classic oscillator model, the vibrational frequency is proportional to the inversed square root of the reduced mass of two bonding atoms. Hence changing the mass of the CN bond can greatly shift its vibrational frequency. (2) The atom at the C-10 position. Our previous study showed replacing O with C or Si atoms can not only shift the absorption maxima but also affect the vibrational frequency of nitrile[18]. (3) The number of expanded rings on xanthene. The substituents on positions other than C-9 may also have influences on the vibrational frequency by remotely modulating the electron density on the CN bond. (4) A sidechain needs to be installed on the dye structure to facilitate subsequent functionalization and convert the dyes into probes with specific targeting abilities. For example, N-hydroxysuccinimide (NHS) is one of the useful functionalizations.

**Synthetic methods for building MARS dyes**. We first synthesized a series of pyronin intermediates which are essential precursors for nitrile group addition. In general, O-cored pyronin rings were built up by condensation reactions between various alkylated 3-aminophenol and corresponding aldehydes with the catalysis of protic acids. Products with intense magenta color could be easily obtained with good yields (Fig. 2a and Supplementary Table 1). The C- or Si- cored pyronins were formed firstly through Friedel-Crafts reactions catalyzed by Lewis acid to connect two scaffolds, and the corresponding intermediates were treated with either protic acid or butyl lithium to finish the ring closures. Next, all the afforded pyronin intermediates were treated with excessive potassium cyanide in a mixed solvent of water and acetonitrile. The electron-deficient C-9 position is readily attacked by cyanide anions, and the reaction completes rapidly within a few minutes. The resulting intermediates are prone to

oxidation and immediately treated with exogenous oxidant FeCl$_3$ to produce desirable 9-cyanopyronins with high yields up to 89% over two steps (Fig. 2b and Supplementary Table 1). This entire one-pot cyano-addition-oxidation process can be finished rapidly within one hour. Furthermore, to facilitate downstream functionalization for targeted labeling, a sidechain with a carboxylic acid terminal was pre-installed on the amino group. As shown in Supplementary Table 1, the addition of the sidechain didn't affect the yields of condensation and cyano addition. The resulting acids were treated with four isotopic potassium cyanides and FeCl$_3$ to afford twelve 9-cyanopyronin carboxyl derivatives and converted to corresponding NHS esters **6a–d**, **7a–d**, and **8a–d** shown in Fig. 2b), which can later be directly used for bioconjugation.

Comparing the absorption spectra of **1–5** with those of corresponding pyronin precursors, the addition of cyano groups at pyronin C-9 positions caused 120 nm red shifts on absorption maximum, resulting in a color change from magenta to dark blue (Supplementary Fig. 2). Similar effects were observed on C and Si-based compounds. The huge change in the dye absorption is a strong manifestation that the nitrile bond is indeed participating in the conjugation system of the dye, which is also required in resonance Raman spectroscopy for a favorable coupling between electronic transition and vibrational transition. In this sense, the absorption spectrum serves a good starting point to verify the electronic resonance condition for the vibrational mode.

Herein we have established robust synthetic methods to build MARS dyes of different core atoms, conjugation ring numbers, and stable isotope substitutions. The new methods greatly improved the synthesis efficiency compared to our prior work in 2017. The previous synthesis routes of MARS dyes highly rely on the commercial availability of pyronin dyes. They either needed commercial Pyronin Y to insert cyanides or Rhodamine 800 to knock out cyano group first and reinsert isotopic cyanides, which resulted in poor overall yields (<9% over 3 steps from Rhodamine 800). We revisited the synthesis of pyronin intermediates and developed simple but robust one-pot condensation to build pyronin cores. All 9-cyanopyronin isotopologues now start with pyronins, directly affording all desired MARS dyes with high yields (up to 89%).

**A quantitative model predicting the structure-spectroscopy relationship of MARS dyes**. Thanks to the developed synthetic methodology, the resulting expanded MARS dyes allowed us to systematically study the mechanisms by which the vibrational frequency is tuned by various structure features. Four notable rules have been revealed concerning the coarse, medium, and fine-tuning mechanisms and their mutual relations, respectively.

The isotope editing offers a classic approach to alter the vibrational spectrum, as demonstrated in our earlier editing of alkyne tags in the non-electronic-resonant cases[25–27]. In the current electronic pre-resonance scenario, isotope editing still works on the nitrile bonds coupled to the core of the dye conjugation system. As expected, isotope doping doesn't change the absorption or emission spectrum. However, the vibrational frequencies were changed dramatically through the modulation of the reduced mass. For instance, four isotopologues compound **6a–d** show CN stretching frequencies at 2238, 2210, 2184, and 2155 cm$^{-1}$ respectively (Fig. 2c). The frequency dropped by ~26–28 cm$^{-1}$ with isotopes added on the bond, which is a significant shift compared to the peak width itself (~10 cm$^{-1}$). Similar trends were observed in series **7a–d** and **8a–d** (Fig. 2c). Hence, isotope editing alters the vibrational frequency in a relatively coarse manner.

Replacing O with C or Si atoms can also affect the vibrational frequency of nitrile, likely due to the change of electron density

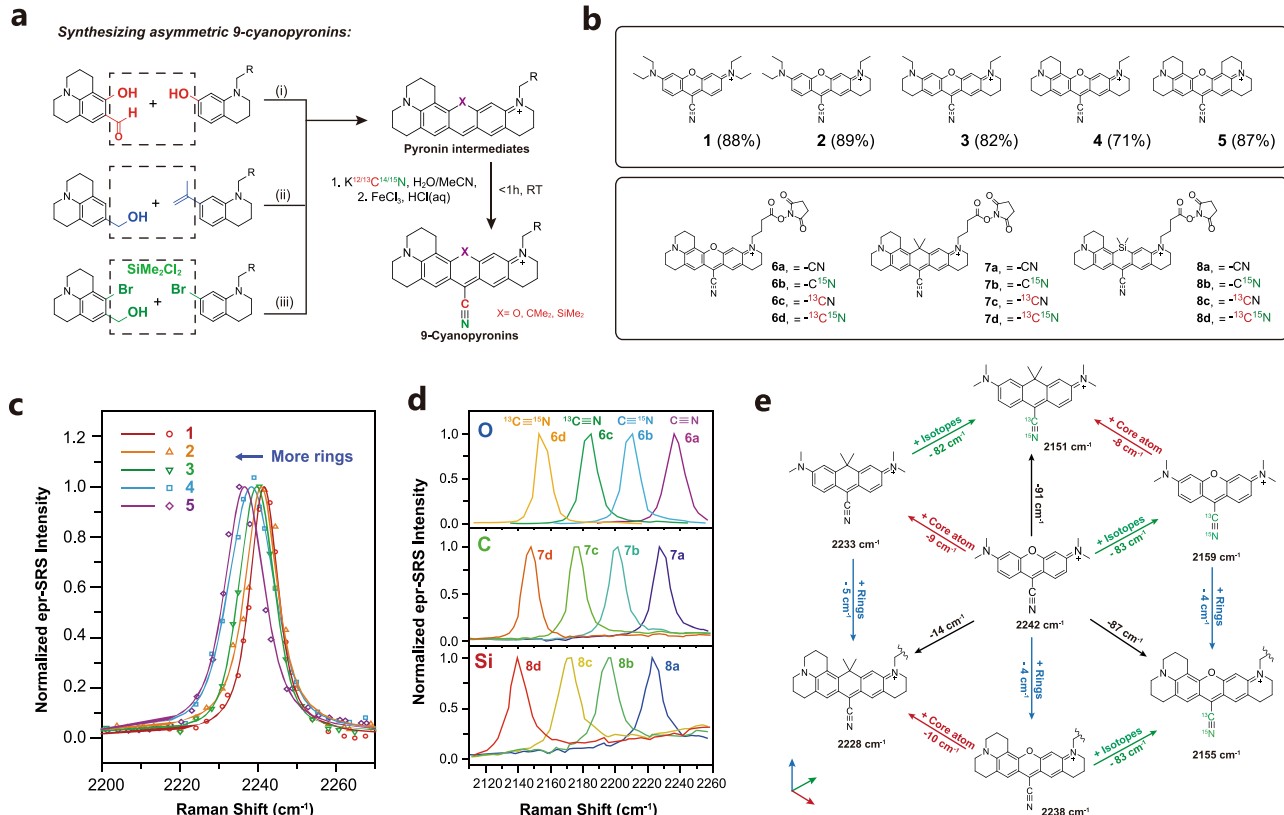

**Fig. 2 Synthesis and spectroscopic study of model MARS probes. a** Synthetic methods to build asymmetric 9-cyanonpyronins. Reaction conditions: i. $H_3PO_4$, 90 °C, overnight. ii. (1) $BCl_3$, DCM, overnight; (2) $H_3PO_4$/PPA, 90 °C, 4 h; (3) $FeCl_3$/HCl(aq), DCM, 2 h. iii. (1) $BCl_3$, DCM, overnight; (2) n-BuLi, THF, −78 °C, overnight. **b** Structures of the model compounds used for spectroscopic study. The first set was used to verify the effect of ring expansion and the second set was used for isotope and core atom effect. **c** Fitted and normalized epr-SRS spectra of compound **1–5** at triple bond region. Curves were Voight fitted to obtain accurate peak positions. **d** Normalized epr-SRS spectra of compound **6a–d**, **7a–d**, and **8a–d** at triple bond region. **e** Additivity and commutativity of three individual effects induced by isotope doping, core atom substitution, and ring expansions. The three axis represents three dimensions of structural modifications, which are mutually orthogonal.

on the nitrile bond. The C-rhodamine and Si-rhodamine were well known to have significant UV–vis redshifts compared with rhodamines[28,29]. This trend was well preserved in 9-cyanopyronin compounds (Supplementary Figs. 3a and 4). The absorption maximum of O-cored MARS dyes **6a–d** was around 690 nm, while the C-cored cyanopyronin showed a 70 nm redshift and Si-cored compound showed an even larger redshift of 100 nm. Replacing core atoms from O- to C- and Si- also results in a substantial shift in the Raman peak of CN. Molecule **6a** with O atom showed CN stretching frequency at 2238 cm$^{-1}$. After replacing the O atom with C core, the peak of molecule **7a** shifted to 2228 cm$^{-1}$. Molecule **8a** bearing a Si core showed an even smaller Raman shift of 2222 cm$^{-1}$. Hence, the center atom alters the vibrational frequency in a relatively moderate manner.

Different numbers of rings on the pyronin can lead to a subtle vibrational shift of CN stretching. With the increasing number of rings, the absorption maxima of compound **1–5** showed gradual redshifts in 5–10 nm increments (Supplementary Fig. 3b). For vibrational spectroscopy, each ring brought an ~1 cm$^{-1}$ decrease in the Raman shifts of CN stretching mode, as shown in Fig. 2e. Hence this is a fine-tuning mechanism relating structure to vibrational spectroscopy.

Close examination of the interplay among the above coarse, moderate and fine tuning mechanisms indicated that these effects could be nearly additive and commutative. For instance, the MARS molecule bearing $^{13}C$ and $^{15}N$ with ring expansions also showed the same difference compared to those without isotope doping, as shown between **MARS2159** and **MARS2242**, and

between **MARS2155** and **MARS2238** in Fig. 2e (Molecules are named according to the Raman shifts measured in DMSO). In addition, the frequency changes brought by center atoms were also observed on isotopologues listed in Fig. 2e. Furthermore, ring expansion and center atom substitution are also two orthogonal tuning rules from each other, as suggested by comparing **MARS2233** with **MARS2242**, and **MARS2228** with **MARS2238**.

Finally we are in a position to integrate the above four rules into a quantitative model to predict vibrational frequencies directly from overall MARS dye structures. As shown in Fig. 3a, a simple linear flow chart with three accumulative layers can be established to estimate the Raman shifts of MARS dyes. In the first layer, the isotope substitution will bring the largest effects on Raman shifts, resulting in 4 nearly evenly distributed Raman peaks spaced by around 26–28 cm$^{-1}$. Next, the core atom, when changing from O to C then Si, sequentially afforded around 8 cm$^{-1}$ changes in peak positions. The last step was determined by the number of 6-membered rings on the chromophore. As each ring gives about 1 cm$^{-1}$ shift, the Raman shifts need to subtract the number of rings prior to resulting in final Raman shifts. To test this model, we combined newly synthesized MARS dyes with compounds demonstrated in the previous report, and plotted all the predicted Raman shifts against the experimentally measured Raman shifts in Fig. 3b (exact wavenumbers were listed in Table 1). The plotted curve showed great linearity and root-mean-square (RMS) of merely 1.12 cm$^{-1}$, indicating the high precision of our model in predicting the vibrational frequencies of nitrile modes in MARS dyes.

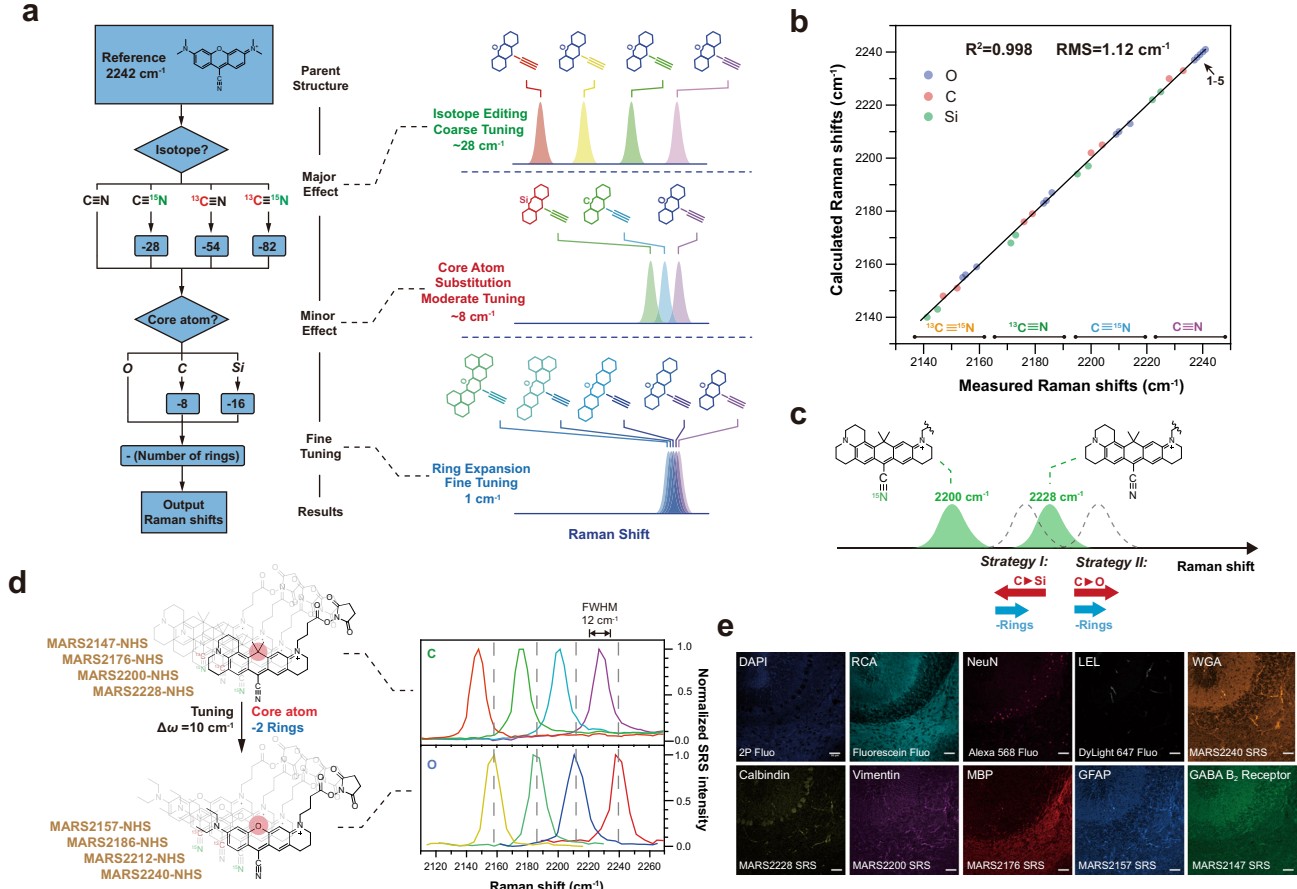

**Fig. 3 Super-multiplexed optical imaging with new MARS probes generated by precise prediction. a** Flow chart used to estimate Raman shifts of MARS dyes. Starting from the reference structure, three layers were used to modify the expected Raman shifts: isotope doping, core atom substitution, and ring expansion. The three effects can be used to precisely control Raman shifts of nitriles. **b** Linear fitting of calculated Raman shifts versus measured Raman shifts. The good linearity and RMS indicate the precision of the propose prediction method. **c** Schematic illustration of two possible strategies to generate new MARS probes that have resolvable frequencies with **MARS2200** and **MARS2228**. Combining core atom substitution and ring expansions can both tune the probe away but the two effects in strategy I unfortunately work in opposite direction, resulting in crosstalk with **MARS2228**. **d** Structures of four new MARS NHS esters designed to be compatible with **MARS2228** series. The measured Raman shifts exactly matched the predicted Raman shifts and the compounds were named accordingly. Top: epr-SRS spectra of **MARS2228** series. Bottom: epr-SRS spectra of **MARS2240** series. The two groups of four peaks are evenly spaced to fit into each other, showing good spectral resolvability. **e** Simultaneous 10-target imaging on mouse cerebellum thin sections. Fluorescence: DAPI (total DNA), RCA (Ricinus Communis Agglutinin I-FTIC), NeuN (rabbit, Alexa568), LEL (Lycopersicon Esculentum lectin Dy-Light 647). SRS: WGA (**MARS2240-WGA**), Calbindin (mouse, **MARS2228**), Vimentin (chicken, **MARS2200**), MBP (myelin basic protein, rat, **MARS2176**), GFAP (glial fibrillary acidic protein, goat, **MARS2157**), GABA B receptor 2 (guinea pig, **MARS2147**). Scale bars, 50 μm. Replicates = 5.

**Rational design of MARS dyes using the prediction model.** The quantitative model established above permits us to precisely predict probes Raman features directly from structures. Conversely, this model would also assist us to design and evaluate new structures when targeting specific frequency range. In our previous study, we have successfully used the four C-cored MARS NHS esters **7a**–**d** to demonstrated multiplexed optical imaging with a high signal-to-noise ratio. Seeking for higher multiplexing, we aimed to leverage the quantitative model to design and synthesize MARS NHS esters that are compatible with the existing four probes.

The four C-cored MARS NHS esters are spaced with nearly uniform intervals of 28 cm$^{-1}$ in the spectral domain, which is the result of coarse tuning from isotope editing. If we generate a set of MARS probes that differ only in isotope editing in the nitrile bond, we should have their peaks spaced in the same intervals of 28 cm$^{-1}$. Considering a typical FWHM of the nitrile peaks for our MARS probes is only 12 cm$^{-1}$, it is possible for the Raman peaks of the previous set and the new set to be interleaved,

thereby separating from each other. The key to this design is how to fill in the spectral interval between **7a** and **7b** by a new dye, as the rest of isotopes will follow the same trend to achieve the interleaving. As shown in Fig. 3c, to fill this interval between **7a** (at 2228 cm$^{-1}$) and **7b** (at 2200 cm$^{-1}$) with minimized spectral crosstalk, the ideal range for this nitrile peak should be 2212–2216 cm$^{-1}$.

According to the quantitative model, we have two potential ways to approach this desired range, as illustrated in Fig. 3c. One can either redshift the peak of **7a** by substituting the core atom from C into Si or blueshift the peak of **7b** from the other side by substituting the core atom from C to O, accompanied by additional fine tuning via ring expansion (Strategy I). However, in the strategy of redshifting **7a**, the ring expansion and the center atom substitution from C into Si work in opposite directions —an overall red shift of <8 cm$^{-1}$ can be expected to give new peaks ranging between 2220 and 2223 cm$^{-1}$, which could not reach our ideal range unfortunately. In contrast, in the strategy of blueshifting **7b**, changing the core atom into O and further

**Table 1 Photophysical properties of newly synthesized MARS model compounds.**

| MARS | Nitrile | Core | Number of ring expansions | Measured Raman shift (cm⁻¹)[a] | Predicted Raman shift (cm⁻¹) | $\lambda_{abs}$ (nm)[b] | RIE[c] |
|---|---|---|---|---|---|---|---|
| 1 | C≡N | O | 0 | 2241 | 2241 | 670 | 75 |
| 2 | | | 1 | 2240 | 2240 | 675 | 86 |
| 3 | | | 2 | 2239 | 2239 | 680 | 93 |
| 4 | | | 3 | 2238 | 2238 | 690 | 108 |
| 5 | | | 4 | 2237 | 2237 | 700 | 120 |
| 6a | C≡N | O | 3 | 2238 | 2238 | 690 | 111 |
| 6b | C≡¹⁵N | | 3 | 2210 | 2210 | | |
| 6c | ¹³C≡N | | 3 | 2184 | 2184 | | |
| 6d | ¹³C≡¹⁵N | | 3 | 2155 | 2156 | | |
| 7a | C≡N | C | 3 | 2228 | 2230 | 760 | 435 |
| 7b | C≡¹⁵N | | 3 | 2200 | 2202 | | |
| 7c | ¹³C≡N | | 3 | 2176 | 2176 | | |
| 7d | ¹³C≡¹⁵N | | 3 | 2147 | 2148 | | |
| 8a | C≡N | Si | 3 | 2222 | 2222 | 790 | 940 |
| 8b | C≡¹⁵N | | 3 | 2195 | 2194 | | |
| 8c | ¹³C≡N | | 3 | 2171 | 2168 | | |
| 8d | ¹³C≡¹⁵N | | 3 | 2141 | 2140 | | |

a, bMeasured in DMSO solution. See supplementary information for spectroscopic data measured in PBS buffer.
cRIE: relative intensity v.s. EdU (5-ethynyl-2′-deoxyuridine) with the same SRS acquisition parameters.

decreasing the ring numbers work synergistically, which could lead to a relatively large blue shift (Strategy II). More quantitatively, our model predicts that changing the core atom into O and removing two of the periphery rings from **7b** (with six 6-member rings) could theoretically generate a nitrile peak around 2212 cm⁻¹, which lies exactly within our desired range of 2212–2216 cm⁻¹.

Guided by this rational design, we performed the synthesis of a set of such MARS probes equipped with O-core atom and only four 6-member rings (Fig. 3d), which are later named according to the measured Raman shifts. The results well matched our prediction: one of the isotopologues indeed exhibits an epr-SRS peak at 2212 cm⁻¹, and the four peaks of the new set are interleaved and can be clearly resolved with those from the previous set of **7a–d**, as shown in Fig. 3d, which was further validated in mixed solutions (Supplementary Fig. 5). We then conjugated them with six designated secondary antibodies. Upon coalition with four fluorescent dye conjugated antibodies, a 10-color one-shot immunostaining was performed on fixed mouse cerebellum thin section (Fig. 3e). The imaging result clearly showed the cell type heterogeneity in cerebellum tissue, and different structures can be unambiguously resolved. In contrast, only 8-color immunostaining (4 NHS ester functionalized MARS probes plus 4 fluorescent probes) was previously possible. The tissue imaging results presented here clearly showed that our newly synthesized NHS ester functionalized MARS probes can collaborate with fluorescence imaging and that the multiplexing capacity can be increased without the need of cyclic labeling and imaging.

**Four-dimensional expansion of functionalized MARS probes with diverse labeling abilities**. The prediction model and synthetic methods allowed us to systematically expand the palette of functionalized MARS probes. Along the spectral tunning dimension, the proposed model facilitated three tunning strategies: major tunning by isotope doping, minor tunning by core atom substitution, and fine tuning by ring expansions. Based on this framework, a 3D library of MARS NHS ester probes was be established (Fig. 4a, detailed structures, and photophysical parameters are listed in Supplementary Fig. 6). Their epr-SRS spectra were acquired and presented in Fig. 4b, covering a wide spectral window of the cell-silent-region, and they are named according to the measured Raman shifts of the conjugated nitrile bonds.

Next, we verified the versatile labeling and imaging capability of the NHS MARS probes (total 20 probes) in cells. Prior to be used in cellular imaging, we tested the MARS probes' performance in aqueous conditions. With the positively charged nature, the MARS NHS esters exhibit good aqueous solubility. The absorption spectra in PBS buffer were shown in Supplementary Fig. 3 and the detailed parameters were listed in Supplementary Table 2. Blueshifts around 15 nm in absorption spectra were observed for all MARS dyes, while slight shifts around 3–6 cm⁻¹ towards larger wavenumbers in SRS spectra were found. Furthermore, the stability of probes was tested in PBS buffer at physiological pH. (Supplementary Fig. 7). The O-cored compounds could preserve 80% effective concentration while the C- and Si- cored compounds could retain 60%. Such difference is in consistency with previous studies on the electrophilicity of pyronins with different core atoms[30–32].

When conjugating with proteins, the probes' degree of labeling can reach 2–4 dye molecules per protein for secondary antibodies. Following the standard immunostaining method, various cellular protein targets can be labeled by secondary antibodies conjugated with MARS probes and subsequently imaged by epr-SRS microscopy. As shown in Fig. 4c, all imaged protein targets showed clear subcellular structure patterns including α-tubulin (microtubule marker) and fibrillarin (nucleoli marker). In addition to NHS esters, our synthetic methods readily allow the construction of MARS probes bearing other functional groups, giving access to many versatile labeling modalities beyond the prior report, thus affording a new fourth dimension to expand MARS library. To save the relatively expensive isotope materials, we presumed the functionalization can be carried out prior to the cyano group addition. The resulting pyronin can later be assigned a nitrile isotope to be compatible with other existing probes. By direct amidation followed by cyanide addition, a group of 10 MARS probes were obtained (Fig. 4d and f), which were aimed for various labeling modalities such as click reaction[33]. Notably, 5 MARS probes were specifically designed for live cell subcellular structure imaging such as mitochondria, lysosome, and lipid structures[34,35].

We then demonstrated imaging with newly functioned MARS probes with diverse labeling modalities. Two clickable MARS

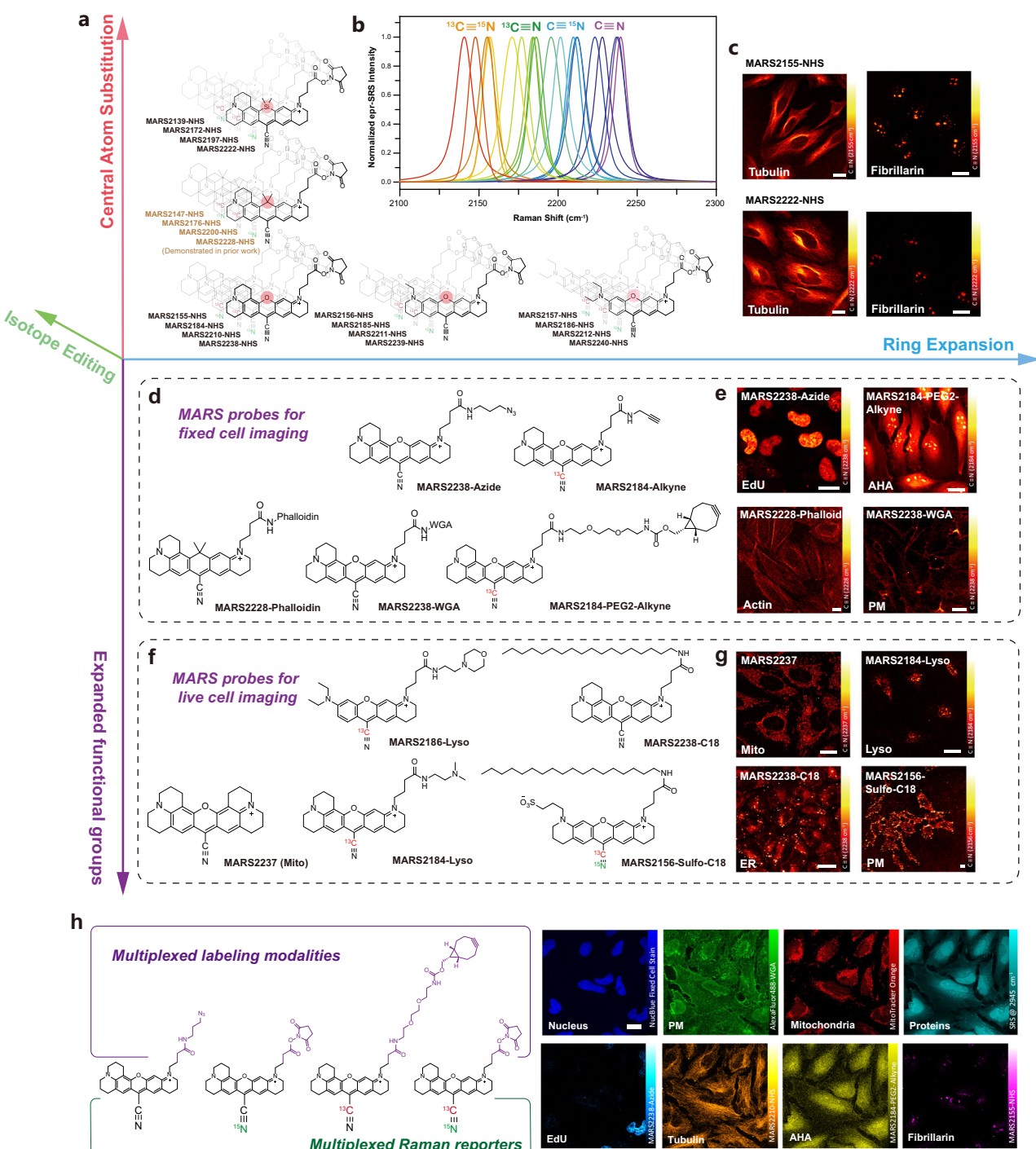

**Fig. 4 Multidimensional tunning strategy to generate spectrally resolvable MARS probe library. a** The probes were rationally engineered along three dimensions: core atom, number of rings, and isotopic substitutions. All probes were named according to fitted nitrile peak positions. **b** Fitted spectra of nitrile stretching modes of 20 MARS NHS esters. 20 peaks can be grouped into 4 sets of isotopologues and clearly resolved. **c** Exemplar images of immunostaining using individual MARS NHS ester probes in fixed cells. Top: Imaging α-tubulin with **MARS2155** conjugated secondary antibody; Imaging fibrillarin with **MARS2155** conjugated secondary antibody. Bottom: α-tubulin imaging with **MARS2222** conjugated secondary antibody; Fibrillarin imaging with **MARS2222** conjugated secondary antibody. Scale bars: 20 μm. Replicates = 3. **d, f** The fourth dimension to expand MARS library. **d** MARS probes developed for fixed sample imaging. **e** Imaging EdU or AHA labeled HeLa cells with **MARS2238-Azide** or **MARS2184-PEG2-Alkyne** through click reaction; Imaging F-actin structure with **MARS2228** conjugated phalloidin; Imaging plasma membrane with **MARS2238** conjugated wheat-germ-agglutinin. Scale bars: 20 μm. Replicates = 3. **f** MARS probes developed for live cell imaging. **g** Imaging mitochondria with **MARS2237** in live cells; Imaging lysosome with **MARS2184-Lyso** in live cells; Imaging intracellular lipids or plasma membrane with C18 derivatized MARS probes. Scale bars: 20 μm. Replicates = 3. **h** 8-color multiplexed cell imaging with 3 fluorescent probes and 4 epr-SRS probes. Fluorescence: NucBlue for nucleus, Alexa-488-WGA for plasma membrane, and MitoTracker Orange for mitochondria. SRS: Label-free at 2945 cm$^{-1}$ for proteins, **MARS2238-Azide** with EdU for newly synthesized DNA, **MARS2210-NHS** labeled antibody for α-tubulin, **MARS2184-PEG2-Alkyne** with AHA for nascent proteins, and **MARS2155-NHS** labeled antibody for fibrillarin. Scale bar: 20 μm. Replicates = 3.

probes with azide or alkyne motifs were utilized to react with 5-ethynyl-2′-deoxyuridine (EdU, newly synthesized DNA marker) or L-azidohomoalanine (AHA, nascent protein marker) for metabolic imaging. The obtained epr-SRS images show the clear nucleus pattern of EdU for newly synthesized DNA, and intracellular distribution for newly synthesized protein, respectively (Fig. 4e, top row). Furthermore, we synthesized a **MARS2228** conjugate with phalloidin, which is a peptide that can tightly bind with F-actin[36]. After HeLa cells were treated with **MARS2228-phalloidin** for 30 min, the clear epr-SRS signal was observed with actin's bundle pattern (Fig. 4e, bottom left). Moreover, **MARS2241** was conjugated with wheat-germ-agglutinin (WGA), a lectin widely used as plasma membrane marker. Imaging results showed signal locates at the plasma membrane predominantly, indicating the probe can target membrane selectively (Fig. 4e, bottom right). Besides applications with fixed cells, we can also demonstrate functionalized MARS probes for live-cell applications, which was not possible with the previous MARS molecules. **MARS2237**, without any side chain derivatization, was used to label mitochondria because of its positive charge and good lipophilicity[37]. Live HeLa cells were incubated with 400 nM **MARS2237** and commercially available MitoTracker Green together. The obtained correlative fluorescence and SRS images show a clear overlap between the fluorescent signal from MitoTracker Green and the SRS signal from **MARS2237** (Fig. 4g and Supplementary Fig. 8). Another organelle, lysosome was also imaged with the MARS probe. We equipped **MARS2238** with a terminal dimethylamino group, whose basicity can be used to target lysosome[35]. The colocalization experiment shows a good overlap between **MARS2238-Lyso** and LysoTracker Orange (Fig. 4g and Supplementary Fig. 6). Moreover, we designed and synthesized a MARS2239 derivative that bears a negative charge and a long lipophilic chain, which is a classic strategy to anchor the probe onto live cell plasma membrane. As a control, **MARS2238-C18** (without the negative charge) shows serious internalization and stains almost all intracellular membrane structures. Due to the negative charge from the sulfonate group, **MARS2156-Sulfo-C18** stops internalization and retains it on the plasma membrane (Fig. 4g).

Finally, we selected four functionalized MARS probes of different types and three commercial fluorescent probes to demonstrate a simultaneous multiplexed optical imaging. Four isotopologues of **MARS2238** with different functionalized conjugations were selected to image fibrillarin via immunostaining, α-tubulin via immunostaining, newly synthesized DNA via click reaction with EdU and nascent protein via click reaction with L-azidohomoalanine, respectively. Fluorescent probes were used to label plasma membrane, nucleus, mitochondria, and actin, respectively. Meanwhile, label-free SRS detection at $CH_3$ vibrational mode (around 2920 cm$^{-1}$) provided the total protein imaging inside cells. As shown in Fig. 4h, all eight targets could be clearly visualized and resolved from each other, demonstrating that the MARS probes of different types are capable of working with each other to break the color barrier, as well as coordinating with conventional fluorescence probes.

## Discussion

Major advances in optical imaging have been largely driven by spectroscopy and imaging probes. In the context of vibrational imaging, the recently developed epr-SRS microscopy significantly boosts the sensitivity of Raman imaging by several orders of magnitude, from the standard millimolar under non-resonance to sub-micromolar level under proper electronic pre-resonance excitation of NIR chromophores[20]. Meanwhile, the proposed MARS palette provides the matching probes to potentially allow for simultaneous imaging of a large number of targets through precisely controlling the sharp vibrational spectrum of the triple bond. While this is a powerful strategy, the development of MARS probe palette is relatively lagging behind, partly due to the different and understudied synthetic chemistry, physical chemistry, and chemical biology of MARS probes from the conventional fluorescence probes. As a result, the structure-spectroscopy relationship of MARS dyes was elusive, and the MARS palette was rather limited in both the absolute number of probes and the types of labeling ability—only 4 NHS ester functionalized probes were synthesized in 2017.

In this work, we developed a robust and efficient methodology to build MARS dyes with different core atoms, conjugation ring numbers, and stable isotope substitutions. Meanwhile, the synthesized MARS dyes allowed us to discover a quantitative model relating the structure features to the vibrational spectroscopy directly. Equipped with this model, we successfully expanded the palette of functionalized MARS probes, in terms of both the number (as many as 30) and the type (more than 10). This expanded palette not only allows increased optical multiplexing capacity than previously available (due to many more NHS ester functionalized probes) but also includes many versatile labeling methods such as click chemistry, peptide vectors, and membrane/organelle targeting, far beyond just the NHS ester functionalization. A clear comparison between the previous MARS molecules and the newly developed palette was summarized in Supplementary Table 3, demonstrating the great significance of chemistry improvements.

As this work is the first systematic study on the chemistry of MARS probes, we envision many further developments beyond what have been demonstrated here. With the facile side chain, MARS probes are ready for exploring more functionalities such as sensing other analytes and micro-environment. Indeed, a large number of interesting sensors have been developed based on xanthenes[29,38,39], the parent structures of MARS probes. In fact, a very recent work reported MARS-derivatized probes for detecting enzyme activities[40]. Hence, we expect MARS probes to make important contribution to the multiplexed sensing applications. Besides, the rapidly growing genetically encoded labeling strategies such as SNAP tag and Halo tag have facilitated site-specific live-cell imaging[3,41]. Considering that epr-SRS microscopy is compatible with live-cell imaging and that MARS dyes are well compatible to equip with these hybrid chemical tags, we anticipate that genetically encoded MARS probes will be built up soon to achieve multicolor imaging with high protein specificity inside live cells.

## Methods

**Synthesis and characterization.** Detailed synthetic routes and characterization data of all newly developed compounds can be found in the supplementary information.

**Instrumentation of epr-SRS microscopy.** SRS imaging was performed with a commercial confocal laser-scanning microscope (FV1200, Olympus) but with customized laser sources. An integrated commercial laser source (picoEmerald, Applied Physics & Electronics, Inc.) was utilized to produce both the Pump and Stokes beams for SRS. The wavelength of Stokes beam was fixed as 1064 nm. The beam was modulated at 8 MHz by an electro-optic modulator (EOM) with >90% modulation depth. The Pump beam was controlled by an optical parametric oscillator (OPO) and had tunable wavelengths within a range from 720 to 990 nm. Both beams had 6 ps pulse width and 80 MHz repetition rate. The two beams were spatially and temporally synchronized and tightly focused onto the sample with a 25× water objective (XLPlan N, 1.05 NA MP, Olympus). After passing through the sample, both transmitted beams were collected by a 1.4 N.A. oil condenser. The Stokes beam was filtered off by a high O.D. bandpass filter (890/220 CARS, Chroma Technology), while the Pump beam was detected by a silicon photodiode (FDS1010, Thorlabs) with a DC voltage of 64 V. The output current was terminated by a 50 Ω terminator and demodulated by a high-frequency lock-in amplifier (HF2LI, Zurich instrument) at 8 MHz frequency. The Pump loss signal at each

pixel was digitized and sent to the FV10 analog channel to generate the images. Unless otherwise mentioned, all cell images were acquired with 4 µs × 10 dwelling time and 320 × 320 pixel number. The on-sample power was 10–25 mW for pump and 25–100 mW for Stokes depending on the detectability of signal from imaging targets.

All fluorescence images were collected on the same microscope platform using CW laser (488, 542, and 635 nm) as excitation source. Two-photon imaging of NucBlue was performed with 780 nm pump beam without Stokes. All images were analyzed with ImageJ.

**Secondary antibody conjugation with MARS NHS esters**. All MARS NHS esters were stored as 20 mM DMSO solutions. Secondary antibody solution (~2 mg/mL, see antibody summary in Supplementary Table 4) was adjusted to pH~8.3 using PBS and sodium bicarbonate. 10 e.q. dye solutions were added to 250 µL antibody solution and the mixture was gently stirred in the dark for 2 hr at room temperature. The conjugated antibodies were purified using gel permeation chromatography (GPC) with Sephadex G-25 (Sigma, G25250). The column (0.5 cm in diameter, 15 cm in length) was filled with swelled Sephadex G-25 and equilibrated with PBS buffer. After loaded with crude antibody solution, the column was eluted with PBS buffer and the first band with light blue (for O-cored MARS) or green (for C and Si cored MARS) was collected. The combined fractions were centrifuged and concentrated with Amicon Ultra centrifugal Filters to reach a final concentration around 1 mg/mL. The resulted antibody solutions were kept in PBS with 5 mM sodium azide and stored at −20 °C. The WGA conjugates were obtained following the same protocol.

**Cell culture, labeling, and sample preparation for epr-SRS imaging**. HeLa cells were cultured with Dulbecco's Modified Eagle's Medium (DMEM, 11965) supplemented with 10% FBS and 1% penicillin-streptomycin. All cell cultures were maintained in a humidified incubator at 37 °C with 5% CO$_2$.

Immunostaining of fixed cell samples. HeLa cells were seeded on a round glass coverslip in a 4-well plate with 500 µL DMEM full growth media for 48 h before reaching a confluence of 70–80%. After removal of media, cells were fixed with 4% PFA for 15 min, blocked with 3% bovine serum albumin (BSA) solution for 30 min, and permeabilized with 0.1-0.5% Triton X-100 solution for 15 min. Primary antibodies (Anti-alpha tubulin monoclonal antibody (mouse, Invitrogen, A11126, 1:50) or Anti-fibrillarin antibody (mouse, Abcam, ab4566, 1:50)) were added to the cells at 1:100 dilution in BSA solution and incubated overnight at 4 °C. After washing off the primary antibody solution, cells were blocked with 10% goat serum for 30 min and treated with MARS labeled secondary antibody with 1:100 or 1:200 dilution in 10% goat serum. The cells were cultured overnight at 4 °C and washed with PBS prior to imaging. The labeling experiments were replicated three times as quality control, and for each sample, at least 3 random field of views (FoVs) were selected to ensure the consistency among cells.

10-color super-multiplexed imaging on fixed mouse cerebellum. PFA fixed 40-µm-thick mouse cerebellum slices were used. Tissues were first blocked with 5% donkey-serum in 0.5% PBST (0.5% Triton X-100 in PBS) for 30 min. After blocking, the solution was replaced by primary antibody solution (desired primary antibodies with 2% donkey serum in 0.5% PBST: Recombinant Alexa Fluor® 568 anti-NeuN antibody (rabbit, Abcam, ab207282), Monoclonal anti-calbindin-D-28K antibody (mouse, Sigma-Aldrich, c9848), Anti-vimentin antibody (chicken, Abcam, ab24525), Anti-Myelin Basic Protein antibody (rat, Abcam, ab7349), Anti-GFAP antibody (goat, Abcam, ab53554), Anti-GABA B Receptor R2 Antibody (guinea pig, Sigma-Aldrich, AB2255)) for 2 days at 4 °C. After primary antibody staining, tissues were washed with 0.5% PBST for 5 min three times followed by blocking solution for 30 min. After blocking, add the secondary antibody solution (desired secondary antibodies with 2% donkey serum in 0.5% PBST), RCA-FITC (ThermoFisher Scientific, L32477), LEL-DyLight 647 (ThermoFisher Scientific, L32472), and **WGA-MARS2242** for 2 days at 4 °C. DAPI solution was stained for 10–15 min after secondary antibody. The tissue was transformed to Superfrost microscopy slide after washing in PBS. Mount and cover slip each slide using a drop of ProLong Gold antifade.

Imaging newly synthesized DNA in fixed HeLa cells by EdU incorporation and **MARS2238-Azide**. HeLa cells were seeded on a round glass coverslip and cultured with DMEM full growth medium until reaching 50% confluence. The medium was then replaced with FBS depleted DMEM for synchronization. After 24 h, the medium was replaced with DMEM full growth medium containing 5 µM EdU and the cells were further incubated for 15 h before fixed with 4% PFA. The fixed cells were permeabilized with 0.5% Triton-X for 10 min. 1 µM **MARS2238-Azide** in Click-iT cell reaction buffer (C10259, Invitrogen) was then added to react with incorporated EdU for 30 min. Cells were washed three times prior to imaging. Three biological replicates and three technical replicates were performed with consistent results.

Imaging newly synthesized protein in fixed HeLa cells by AHA incorporation and **MARS2184-PEG2-Alkyne**. HeLa cells were seeded on a round glass coverslip and cultured with DMEM full growth medium until reaching 50% confluence. The medium was then replaced with methionine-deficient DMEM containing 1 mM AHA. After incubating for 18 h, cells were fixed with 4% PFA and permeabilized with 1% Triton-X for 10 min. 5 µM **MARS2184-PEG2-Alkyne** was added to the PBS buffer and hatched for 1 h to complete copper-free click reaction. Cells were

washed three times prior to imaging. Three biological replicates and three technical replicates were performed with consistent results.

Imaging F-actin in fixed HeLa cells with **MARS2228-Phalloidin**. **MARS2228-NHS ester** (0.1 mM) was incubated with phalloidin amine (0.1 mM, AAT Bioquest 5302) in basic PBS buffer (pH = 8.3) for 1 h. The reaction was monitored and confirmed by mass spectroscopy. The resulting mixture was directly used for fixed and permeabilized HeLa cells labeling with 100× dilution. Cells were washed three times prior to imaging. three technical replicates were performed with consistent results.

Imaging subcellular structures of live HeLa cells with organelle-targeting MARS probes. HeLa cells were incubated with DMEM medium until reaching 70% confluence. Designated MARS probes: **MARS2237** (500 nM), **MARS2184-Lyso** (5 µM), **MARS2238-C18** (5 µM), **MARS2156-Sulfo-C18** (2 µM) were added into culture media and hatched for 30 min. Cells were washed with PBS and directly placed on glass slide for imaging. To help with dissolving, **MARS2238-C18** was mixed with Pluoronic F-127 with 1:1 ratio before adding to cells. The labeling was replicated three times as quality control, and for each sample, at least three random field of views (FoVs) were selected to ensure the consistency among cells.

Eight color epr-SRS and fluorescence tandem imaging of fixed HeLa cells. HeLa cells were seeded on round coverslip and cultured to 70% confluence. The medium was replaced with methionine free DMEM containing 5 µM EdU and 100 µM AHA. The cells were incubated for 2 h followed by the addition of 200 nM MitoTracker Orange and 10 µg/mL WGA-Alexa 488 conjugate. Cells were further hatched for 30 min before fixed with 4% PFA for 15 min and permeabilized with 2% Triton-X for 10 min. 3% BSA was used to block cells for 30 min. The treated HeLa cells were subject to primary antibody labeling (anti-fibrillarin from rabbit, (Invitrogen PA5-29801, 1:50), and anti-tubulin from mouse, (Invitrogen A11126, 1:25)) in BSA overnight, then secondary antibody labeling (MARS2155-goat-anti-rabbit and MARS2210-goat-anti-mouse) in goat serum for 6 h. After washing with PBS, click reactions were performed on cells. 1 µM **MARS2238-Azide** in Click-iT buffer was added and reacted for 30 min. The sample was thoroughly washed before the addition of 1 µM **MARS2184-PEG2-Alkyne**. After incubation for 1 h, cells were labeled with NucBlue Fixed Cell ReadyProbes reagent (Invitrogen, R37606) for 10 min. Cells were washed again prior to imaging. Three biological replicates and three technical replicates were performed with consistent results.

**Reporting summary**. Further information on research design is available in the Nature Research Reporting Summary linked to this article.

## Data availability
The data that support the findings of this study are all provided with this paper in supplementary information or source data files. Source data are provided with this paper.

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

## Acknowledgements
We thank Professor B. Stockwell and the group for the help in chemical synthesis and purification, B. Fowler and J. Decatur for molecule structural characterizations. W.M. acknowledges support of R01 GM128214 and R01 GM132860 from National Institute of Health, and National Science Foundation (1904684).

## Author contributions
The manuscript was written through contributions of all authors. Y.M., L.S., F.H. and W.M. conceived the concept. Y.M. and N.Q. carried out the chemical synthesis, characterization, spectroscopy, and cellular imaging experiments. L.S. carried out the super-multiplexed tissue imaging. Y.M. and W.M. wrote the manuscript with input from all authors.

## Competing interests
The authors declare no competing interests.
