## [Peer Review File · Nature Communications]

REVIEWER COMMENTS

Reviewer #1 (Remarks to the Author):

The manuscript from Min et al. established robust methods to synthesize/design 9-cyanopyronins (MARS dyes) with different core atoms, conjugation ring numbers, and stable isotope substitutions for super-multiplexed optical imaging. It is quite interesting that the authors established a model to predict vibrational frequencies of MARS dyes, which would be beneficial for efficient development of new Raman probes and fine tuning of vibrational frequencies. Since this manuscript is thought to be a great advance over their previous work in terms of number/diversity/practicality of probes, together with the recent demand for multiplexed imaging, this reviewer is highly supportive for the publication of this manuscript in Nature Communications after the following revisions.

1) The information of relative Raman intensity of the newly developed MARS dyes would be informative for readers, as the Raman cross-section can vary depending on the absorption wavelengths according to their previous report (Nature 2017).

2) Absorption spectra and vibrational frequencies seem to be measured in DMSO according to SI. It would be better to mention that these properties were measured in DMSO also in the Table/Figure legends. Also, this reviewer thinks it would be informative for readers if the authors can provide absorption spectra and vibrational frequencies of these dyes in aqueous buffer as most of these dyes are supposed to be used in the aqueous buffer for cellular imaging.

3) Fine tuning of vibrational frequency with core atom and conjugation ring number in Figure 3c-d is so splendid as the vibrational frequency of new set were interleaved beautifully among the previous set. It would be interesting if the authors can demonstrate that the signal of these eight dyes in Figure 3d can be detected even when mixed. Also, I and readers would expect multiplexed imaging by using these 8 dyes (for example, in Figure 3e), but the authors demonstrated imaging with 6 dyes together with 4 fluorescent dyes. Was the multiplexed imaging with these 8 dyes practically difficult to perform?

4) Baseline of Raman spectra of Si-cored derivatives in Figure 2d seems to be higher compared to other derivatives (O-cored, C-cored). Can the authors provide some possible explanation for this high background?

5) It would also be nice to add the information about stability of the newly developed dyes in the aqueous buffer, since it has been reported that some of 9-CN pyronins are not so stable in aqueous buffer.

6) This reviewer is not convinced with the statement of "Interestingly, the substituents at C-4 and C-5 positions (i.e., the upper two carbon atoms) induced slightly larger shifts than those at C-1 and C-8 positions (i.e., the bottom two carbon atoms)". I am wondering whether this statement would not be concluded unless the authors developed a derivative with ring fusion at C-4 and C-5 positions (without ring fusion at C-1 and C-8 positions) and evaluated vibrational frequency to compare that of a derivative with ring fusion at C-1 and C-8 positions, as the effect of these substituents would not be independent.

Minor points;

1) Figure S3a and Figure S3b should be replaced with each other to cite the appropriate figures in the main text (page 4).

2) Figure S3a; absorption spectra seem to be mis-labeled, as the absorption wavelengths would be extended with increased number of ring fusion and derivative 1 would show shortest wavelength.

3) Figure S4; I am wondering why fluorescence spectra of compound 6a is missing, while appeared in the absorption spectra of Figure S3.

4) Figure S2; Synthetic scheme of compound 14 was described, but there seems to be no description about this compound in the main text.

5) Figure 3a, Figure 4a; chemical structures with stable isotope labelling are faint and hard to see, which may need some improvements.

Reviewer #2 (Remarks to the Author):

The paper by Maio et al. is another important contribution to development of Raman probes by the Min group.

This paper is primarily focused on synthetic innovations, but analyzing the many compounds synthesized allowed them to note effects of molecular composition changes on the peak locations for CN stretch. They found that the effects were additive, allowing them to rationally design molecules with desired vibrational frequency.

Overall, the paper is well-written and clear, and describes what seems to me to be an impressively flexible synthetic approach. However...

In a sense the paper is incremental as this group has already produced many MARS probes, some with intrinsic target affinity, and some conjugated with affinity enhancers. Also - while the quantitative prediction of vibrational frequency is very nice, the principles involved (e.g. that isotopic changes induce notable frequency shifts) are well known. So, while the model is interesting, it is not particularly surprising, and it is perhaps a bit parochial in its utility.

Given this, I wonder what the broad interest would be for this paper.

- Is it simply the many new probes? Was the previous set sufficiently small to induce functional limitations on the information they could be used to convey? (Was there some situation in which it was just not possible to get enough information from the previously available MARS probes?)
- Is it the new synthetic routes? (I am not suited to judge this)
- Did the new synthetic routes enable some functionality that was not available previously?

Almost parenthetically, at the end of the last full paragraph on page 6, the authors write that for the first time, MARS probes were designed for live cellular imaging. Was there something about the new synthetic routes that facilitated this? Was it not possible to synthesize such probes previously?

While the paper is impressive, and I am certainly not opposed to it being published in Nature Communications, I think it would benefit from a bit more clarity on what functionality was not previously available, and what was not possible (other than optimally filling a narrow vibrational spectral range) that now is.

Response to Reviewer 1:

The manuscript from Min et al. established robust methods to synthesize/design 9-cyanopyronins (MARS dyes) with different core atoms, conjugation ring numbers, and stable isotope substitutions for super-multiplexed optical imaging. It is quite interesting that the authors established a model to predict vibrational frequencies of MARS dyes, which would be beneficial for efficient development of new Raman probes and fine tuning of vibrational frequencies. Since this manuscript is thought to be a great advance over their previous work in terms of number/diversity/practicality of probes, together with the recent demand for multiplexed imaging, this reviewer is highly supportive for the publication of this manuscript in Nature Communications after the following revisions.

First of all, we would like to thank this reviewer for his/her support of our work! Following his/her suggestions, we have included new data in the Supporting Information.

1) The information of relative Raman intensity of the newly developed MARS dyes would be informative for readers, as the Raman cross-section can vary depending on the absorption wavelengths according to their previous report (Nature 2017).

We agree with the reviewer that such information will be informative and helpful for the readers. Following this suggestion, we've added the RIEs (relative intensity v.s. EdU, a more commonly used and intuitive description than absolute cross sections) to **Table 1** (measured in DMSO) and **Supplementary Table S2** (measured in PBS). Copies of the updated tables are attached here:

Table 1. Photophysical properties of newly synthesized MARS model compounds in DMSO.

MARS	Nitrile	Core	Number of ring expansions	Measured Raman shift (cm ⁻¹) ^a	Predicted Raman shift (cm ⁻¹)	λ_{abs} (nm) ^b	RIE ^c
1		O	0	2241	2241	670	75
2		O	1	2240	2240	675	86
3	C≡N	O	2	2239	2239	680	93
4		O	3	2238	2238	690	108
5		O	4	2237	2237	700	120
6a	C≡N			2238	2238		
6b	C≡ ¹⁵ N			2210	2210		
6c	¹³ C≡N	O	3	2184	2184	690	111
6d	¹³ C≡ ¹⁵ N			2155	2156		
7a	C≡N			2228	2230		
7b	C≡ ¹⁵ N			2200	2202	760	435
7c	¹³ C≡N	C	3	2176	2176		
7d	¹³ C≡ ¹⁵ N			2147	2148		
8a	C≡N			2222	2222		
8b	C≡ ¹⁵ N			2195	2194		
8c	¹³ C≡N	Si	3	2171	2168	790	940
8a	¹³ C≡ ¹⁵ N			2141	2140		

a, b: Measured in DMSO solution. See supplementary information for spectroscopic data measured in PBS buffer.
c: RIE: relative intensity v.s. EdU (5-ethynyl-2'-deoxyuridine) with same SRS acquisition parameters.

Table S2. Photophysical properties of newly synthesized MARS model compounds in PBS buffer.
MARS	Nitrile	Core	Number of ring expansions	Measured Raman shift (cm ⁻¹) ^a	λ_{abs} (nm) ^b	RIE ^c
1		O	0	2247	655	60
2		O	1	2246	660	80
3	C≡N	O	2	2244	665	87
4		O	3	2242	675	90
5		O	4	2240	685	103
6a (MARS2238-NHS)	C≡N	O	3	2239	675	92
MARS2239-NHS	C≡N	O	2	2244	665	87
MARS2240-NHS	C≡N	O	1	2246	660	67
7a (MARS2228-NHS)	C≡N	C	3	2234	740	342
8a (MARS2222-NHS)	C≡N	Si	3	2225	770	683

a,b: Measured in PBS aqueous solution.

c: RIE: relative intensity v.s. EdU (5-ethynyl-2'-deoxyuridine) with same SRS acquisition parameters.

2) Absorption spectra and vibrational frequencies seem to be measured in DMSO according to SI. It would be better to mention that these properties were measured in DMSO also in the Table/Figure legends. Also, this reviewer thinks it would be informative for readers if the authors can provide absorption spectra and vibrational frequencies of these dyes in aqueous buffer as most of these dyes are supposed to be used in the aqueous buffer for cellular imaging.

We have added notes in the caption that the spectral data were taken in DMSO solution. We agree that adding spectral data in aqueous conditions will be more informative. The absorption spectra in PBS were added into **Supplementary Figure S3** below. Notably, with the increased number of rings, a significant rise in the shoulder peak absorbance was observed. This can be attributed to the formation of dimers in aqueous environment. The addition of rings increases the rigidity of pyronin and forwards the dimerization equilibrium.¹ Additionally, most 9-cyanopyronins show ~15 nm shorter absorption profile in PBS than those in DMSO. It resulted in the decrease of electronic pre-resonance effect and SRS cross sections (~30%). We listed the RIE data obtained in PBS for reference (see Table S2 above).

UV-Vis absorption in DMSO:

UV-Vis absorption in PBS:

Supplementary Figure S3 UV-Vis absorption spectra of newly synthesized model MARS compounds. (a) Normalized absorption spectra of **1-5** measured in DMSO (10 μM) showing the gradual shift in maximum wavelength. (b) Normalized absorption spectra of O-, C-, and Si- cored MARS dyes. (c) and (d): corresponding absorption spectra measured in PBS buffer (10 μM).

3) Fine tuning of vibrational frequency with core atom and conjugation ring number in Figure 3c-d is so splendid as the vibrational frequency of new set were interleaved beautifully among the previous set. It would be interesting if the authors can demonstrate that the signal of these eight dyes in Figure 3d can be detected even when mixed. Also, I and readers would expect multiplexed imaging by using these 8 dyes (for example, in Figure 3e), but the authors demonstrated imaging with 6 dyes together with 4 fluorescent dyes. Was the multiplexed imaging with these 8 dyes practically difficult to perform?

We performed the spectral unmixing experiment using two representative probes: MARS2240-NHS and MARS2228-NHS. We chose to demonstrate unmixing of these two probes because their spectral separation ($\sim 12 \text{ cm}^{-1}$) is among the tightest of all probe pairs. MARS2240-NHS and MARS2228-NHS probes were mixed in-vitro to obtain a mixture solution where two SRS peaks show approximately identical intensities. The spectra can be readily split into two peaks after simple unmixing. The two separated peaks show the exact Raman shifts as each individual probes, proving the robustness of the unmixing. The results are shown here:

Supplementary Figure S5. In-vitro unmixing of **MARS2228-NHS** and **MARS2240-NHS**. Two probes were diluted with DMSO to obtain a mixture displaying similar SRS intensities. The deconvolution was performed based on Voigt multi-peak fitting. Two separated peaks show exact the same Raman shifts as two probe components.

The ultimate goal of this dye palette is no doubt utilizing all 8 epr-SRS probes. But in practice, the maximum number was limited by the number of commercially available secondary antibodies that are from same species. To perform the 10-color immunostaining demonstrated in our manuscript, we have used donkey-anti-rabbit/mouse/chicken/rat/goat/guinea pig secondary antibodies, which almost covered all commercial sources that we could find. While direct conjugation with primary antibodies can overcome this species limit, primary antibodies are much more expensive than secondary antibodies, which gets even worse as the number of channels goes up for super-multiplexing. We plan to pursue this direction in the future.

4) Baseline of Raman spectra of Si-cored derivatives in Figure 2d seems to be higher compared to other derivatives (O-cored, C-cored). Can the authors provide some possible explanation for this high background?

The Si-cored derivatives have the longest absorption wavelengths (790 nm) among all dyes, which are the closest to our pump laser wavelength (~860 nm). When the energy difference becomes smaller (i.e., approaching rigorous electronic resonance), the background will start to rise. At rigorous resonance, the Raman features are eventually overwhelmed by the broad background, as we have studied previously in electronic resonance SRS spectroscopy.² This is why probe absorption in the electronic pre-resonance region is favored for boosting the Raman signal while retaining high contrast over electronic background.³ This background originates from electronic responses of the molecules such as two-photon absorption, stimulated emission, excited-state absorption, and ground-state depletion, as well as other four-wave-mixing and pump-probe processes.⁴⁻⁵

In our imaging practices, we take off-resonance images by detuning the pump wavelength away from the triple bond region to obtain the background signal, which were subtracted from the original images.

5) It would also be nice to add the information about stability of the newly developed dyes in the aqueous buffer, since it has been reported that some of 9-CN pyronins are not so stable in aqueous buffer.

We thank the reviewer for this suggestion. To test the stability of newly synthesized dyes, we measured the absorption spectra of three MARS NHS esters with different core atoms over a prolonged storage period in PBS buffer. The spectra were obtained every 24 h over one week, and the results are shown here:

Supplementary Figure S7. Stability of 9-cyanopyronin probes in aqueous conditions. 10 μ M of three representative 9-cyanopyronin probes in PBS buffer were stored in dark at room temperature and the absorbance was monitored every 24 h over one week.

The O-cored 9-cyanopyronin presents much better stability than C-cored and Si-cored pyronin. This is not surprising according to the reported computational studies. The pyronins are sensitive to nucleophiles such as thiols, amines, and hydroxides.⁶⁻⁷ Compared to the O-cored pyronins, the C-cored and Si-cored pyronins bear much lower LUMOs indicating they are more electrophilic in thermodynamics.⁸⁻⁹

The results suggest that 9-cyanopyronins can retain at least 60% effective concentrations in PBS over one week storage at room temperature. Considering most bio-conjugation and cell/tissue labeling experiments require reaction time from a few hours to a few days, we believe the stability is not a critical concern for imaging practices.

6) This reviewer is not convinced with the statement of “Interestingly, the substituents at C-4 and C-5 positions (i.e., the upper two carbon atoms) induced slightly larger shifts than those at C-1 and C-8 positions (i.e., the bottom two carbon atoms)”. I am wondering whether this statement would not be concluded unless the authors developed a derivative with ring fusion at C-4 and C-5 positions (without ring fusion at C-1 and C-8 positions) and evaluated vibrational frequency to compare that of a derivative with ring fusion at C-1 and C-8 positions, as the effect of these substituents would not be independent.

We agree with the reviewer that this statement would need additional evidence from a C-4 and C-5 only derivative. As this observation is not a key point and it's not critical to the following studies, we decided to remove this point in our revision. We thank the reviewer for the correction.

Minor points:

1) Figure S3a and Figure S3b should be replaced with each other to cite the appropriate figures in the main text (page 4).

2) Figure S3a; absorption spectra seem to be mis-labeled, as the absorption wavelengths would be extended with increased number of ring fusion and derivative 1 would show shortest wavelength.

We thank the reviewer for the detailed corrections. **Figure S3** has been revised and all legends were corrected.

3) Figure S4; I am wondering why fluorescence spectra of compound **6a** is missing, while appeared in the absorption spectra of Figure S3.

The fluorescence spectra of compound **6a** is almost identical to that of compound **4** as they share the same dye scaffold. For the best vision clarity, we didn't show the spectra of **6a**. As the reviewer suggested, we added it into **Figure S4** now. In addition, we took all emission spectra with a new spectrometer, which has better sensitivity in infrared region. The analyte concentrations were reduced, and the optical path were shortened to minimize self-absorption effect. The figure has been reproduced now:

Supplementary Figure S4. Normalized fluorescence emission spectra of representative 9-cyanopyronin dyes in DMSO.

4) Figure S2; Synthetic scheme of compound 14 was described, but there seems to be no description about this compound in the main text.

The compound **14** was to test the generality of the condensation reaction to build pyronin rings. Its spectroscopic properties were also studied. Its electronic and Raman features are almost identical to that of MARS2239 (O-cored with 2 six-membered rings):

To keep the consistency among the palette, we removed this compound from the main text during manuscript writing but preserved it in the synthesis scheme.

5) Figure 3a, Figure 4a; chemical structures with stable isotope labelling are faint and hard to see, which may need some improvements.

We are sorry for the bad image qualities after compression. We rescaled the figures and optimized the resolution.

Response to Reviewer 2:

The paper by Maio et al. is another important contribution to development of Raman probes by the Min group.

This paper is primarily focused on synthetic innovations, but analyzing the many compounds synthesized allowed them to note effects of molecular composition changes on the peak locations for CN stretch. They found that the effects were additive, allowing them to rationally design molecules with desired vibrational frequency.

Overall, the paper is well-written and clear, and describes what seems to me to be an impressively flexible synthetic approach.

We thank the reviewer for the nice summary and strong appreciation of our work! To address the reviewer's subsequent suggestions and critiques, we have made substantial clarifications in the revision which should make the advance clear to the readers. On one hand, we agree that the current paper follows the general principle laid out in our prior work (*Nature* 2017). On the other hand, we hope that the clarifications below can persuade the reviewer that the current paper made the real imaging applications possible. **For a quick summary, please see the newly added Supplementary Table S3 (listed at the end of this response letter) which makes detailed comparison between previously reported MARS molecules and newly developed molecules.** The field of fluorescence microscopy gives an excellent example, as those novel fluorescent probes are driving the innovations forward.¹⁰ In particular, the syntheses of new probes, studies on photochemical properties, and the optimization of bio-conjugation methods are indispensable behind the prosperity of optical microscopy. In this sense, we believe the current paper (after revision) represents a significant advance in the field of Raman probe development.

However... In a sense the paper is incremental as this group has already produced many MARS probes, some with intrinsic target affinity, and some conjugated with affinity enhancers. Also - while the quantitative prediction of vibrational frequency is very nice, the principle involved (e.g., that isotopic changes induce notable frequency shifts) are well known. So, while the model is interesting, it is not particularly surprising, and it is perhaps a bit parochial in its utility.

In our 2017 work, MARS was presented more as proof-of-principle that the 9-cyanopyronin can be engineered to have tunable Raman shifts. But most of the previous MARS compounds have no functionality or targeting affinity, because we lacked robust and efficient methods to synthesize MARS dyes from common starting materials. As a result, a lot of probes used in cellular imaging demonstrations in 2017 were commercially available NIR fluorescent probes (i.e., not MARS probes) whose double bond stretching modes were used as imaging contrast. It is in this current work that we explored the MARS probes' functionalities and achieved

what the reviewer mentioned as “some with intrinsic target affinity, and some conjugated with affinity enhancers”.

Regarding the quantitative model, we agree that the isotope effects are well known. But there is a lack of quantitative measurements of such effects in specific molecules. The nontrivial thing is that the vibrational frequencies are not solely dictated by the mass of two bonding atoms but also involving the surrounding atoms.¹¹ The total mass on each side of the bonds must be treated as a whole to consider, so it is necessary to examine the isotope effects in particular structures. More importantly, the effects from core atoms (C vs. O vs. Si) and ring expansions are not explored in the context of Raman dye development to the best of our knowledge. We quantified these effects and classified them into three tuning strategies at different levels. Moreover, the analysis of the many compounds synthesized reveals that the effects were additive, allowing us to build a model with predictive power. We did not expect such a simple additive effect ahead of time. We hope to convince the reviewer that this quantitative model is significant and inspiring although may not be totally “surprising”.

Given this, I wonder what the broad interest would be for this paper.

- Is it simply the many new probes? Was the previous set sufficiently small to induce functional limitations on the information they could be used to convey? (Was there some situation in which it was just not possible to get enough information from the previously available MARS probes?)

- Is it the new synthetic routes? (I am not suited to judge this)

- Did the new synthetic routes enable some functionality that was not available previously?

Almost parenthetically, at the end of the last full paragraph on page 6, the authors write that for the first time, MARS probes were designed for live cellular imaging. Was there something about the new synthetic routes that facilitated this? Was it not possible to synthesize such probes previously?

We appreciate these questions from the reviewer. Motivated by these, we have added more clarifications to the revision. The current manuscript involves diverse topics ranging from probe design, synthesis, spectroscopic characterization, functionalization and imaging demonstrations. Hence we believe it would elicit broad interest. Again, we sincerely thank the reviewer for the comments and suggestions.

There are several major limitations in the previous probe set reported in 2017. Only the isotope effect and core-atom effect were demonstrated. Due to the lack of synthetic methods, MARS dyes with different ring expansions were not obtained. Thus, this important tuning dimension was only proposed but not experimentally proven. There were no sidechains for functionalization. In the 2017 work, because of the reliance on commercial pyronin dyes, the proposed MARS dyes had to be symmetric in structure and consequentially had no functionalizable sidechains for targeting capability – in other words, they were just dyes but not imaging probes yet. In fact, there were only 4 NHS ester functionalized probes (all in the same type) in the original MARS palette. Therefore, the imaging applications were incomplete with the previous probe set. It initiated a promising direction but its full potential was far from being exploited.

The current work not only fills up the missing information with many new probes but also provides a systematic study over 9-cyanopyronins: To **create**: synthetic methods; To **understand**: spectroscopic studies; To **utilize**: turning dyes to probes with versatile functions. The new synthesis routes broadened the number, type, and functionality of the available 9-cyanopyronins. Without the innovations in synthesis, the following studies would surely be hindered. The newly developed synthetic methods contribute to the probe palette expansion in the following ways:

- 1) The introduction of isotopic nitriles. Previous O-cored MARS dyes relied on commercially available 9-cyanopyronin dyes (e.g., Rhodamine 800), or pyronins (e.g., Pyronin Y). To obtain the isotope labeled Rhodamine 800, the synthesis had to start from knocking out the nitrile group from Rhodamine 800, which is extremely inefficient. Also, the available pyronins are scarce. In the current work we have proposed a robust reaction to build asymmetric pyronin intermediates. The method greatly saved time and effort to introduce isotopic cyanides into pyronins.

Prior work:

This work:

- 2) The introduction of terminal sidechains. Most commonly used fluorescein-/rhodamine-based probes (e.g., Alexa Fluor series, ATTO series) attach C-9 position with phenyl rings, where sidechains can be installed easily. However, as this position is occupied by nitrile in MARS probe, it requires new chemistry to install the sidechains. Our synthetic routes solved the problem for pyronins with variable core atoms which require distinct synthetic designs.
- 3) The incorporation of targeting/labeling vectors. The addition of functional groups is not plain work by amide bond formation. When we tried to link the terminal carboxyl of MARS with the amino groups, the MARS molecules seemed sensitive to amine with the existence of other strong bases, leading to bad yields. We

did mechanism study and optimized the synthetic route: the addition of functional groups need to be prior to addition of nitriles, because the pyronin intermediates are not vulnerable to amines.

Proposed mechanism:

We hope our explanations can clarify the significance of such improvements in synthetic methods. We also wish the reviewer can understand that we could not present the entire story in details due to the length concern. Nevertheless, we have modified the introduction and the discussion sections to strengthen the comparison between the previous and current probe set.

While the paper is impressive, and I am certainly not opposed to it being published in *Nature Communications*, I think it would benefit from a bit more clarity on what functionality was not previously available, and what was not possible (other than optimally filling a narrow vibrational spectral range) that now is.

We thank the reviewer for his/her appreciation. To provide more clarity, the newly added **Supplementary Table S3** makes clear comparison between the previously reported MARS dyes/probes and the new dyes/probes in terms of chemistry improvement. The first category is what was done before and intact in current work. The second row refers to what was demonstrated before, but the synthetic routes have been significantly optimized to make them easily available. The third row refers to what the reviewer said “not possible before”, but now have been achieved with new synthetic methods. We wish this new Table can convince the reviewer of the advance made in the current manuscript.

Supplementary Table S3. Comparison between previously reported MARS molecules with newly developed molecules.

	Symmetry	Sidechain	Functionality	Examples
Previously reported	Symmetric	No	No	 X= O, CMe ₂ , SiMe ₂
Previously reported but with improved chemistry	Symmetric	No	No		Asymmetric	Yes	NHS-ester	Newly developed probes	Asymmetric	No	No		Asymmetric	Yes	NHS-ester	 X= O, SiMe ₂
	Asymmetric	Yes	Click chemistry		Asymmetric	Yes	Organelle targeting		Asymmetric	Yes	Cell skeleton targeting	 Phalloidin
Asymmetric	Yes	Lipid structure targeting		

References

1. Sekiguchi, K.; Yamaguchi, S.; Tahara, T., Formation and dissociation of rhodamine 800 dimers in water: steady-state and ultrafast spectroscopic study. *J. Phys. Chem. A* **2006**, *110* (8), 2601-6.
2. Shi, L.; Xiong, H.; Shen, Y.; Long, R.; Wei, L.; Min, W., Electronic Resonant Stimulated Raman Scattering Micro-Spectroscopy. *J. Phys. Chem. B* **2018**, *122* (39), 9218-9224.
3. Wei, L.; Chen, Z.; Shi, L.; Long, R.; Anzalone, A. V.; Zhang, L.; Hu, F.; Yuste, R.; Cornish, V. W.; Min, W., Super-multiplex vibrational imaging. *Nature* **2017**.
4. McCamant, D. W.; Kukura, P.; Mathies, R. A., Femtosecond broadband stimulated Raman: a new approach for high-performance vibrational spectroscopy. *Appl. Spectrosc.* **2003**, *57* (11), 1317-23.
5. Zhang, D.; Wang, P.; Slipchenko, M. N.; Cheng, J. X., Fast vibrational imaging of single cells and tissues by stimulated Raman scattering microscopy. *Acc. Chem. Res.* **2014**, *47* (8), 2282-90.
6. Uno, S. N.; Kamiya, M.; Yoshihara, T.; Sugawara, K.; Okabe, K.; Tarhan, M. C.; Fujita, H.; Funatsu, T.; Okada, Y.; Tobita, S.; Urano, Y., A spontaneously blinking fluorophore based on intramolecular spirocyclization for live-cell super-resolution imaging. *Nat Chem* **2014**, *6* (8), 681-9.
7. Umezawa, K.; Yoshida, M.; Kamiya, M.; Yamasoba, T.; Urano, Y., Rational design of reversible fluorescent probes for live-cell imaging and quantification of fast glutathione dynamics. *Nat Chem* **2017**, *9* (3), 279-286.
8. Kushida, Y.; Nagano, T.; Hanaoka, K., Silicon-substituted xanthene dyes and their applications in bioimaging. *Analyst* **2015**, *140* (3), 685-695.
9. Liu, J.; Sun, Y. Q.; Zhang, H.; Shi, H.; Shi, Y.; Guo, W., Sulfone-Rhodamines: A New Class of Near-Infrared Fluorescent Dyes for Bioimaging. *ACS Appl Mater Interfaces* **2016**, *8* (35), 22953-62.
10. Lavis, L. D., Teaching Old Dyes New Tricks: Biological Probes Built from Fluoresceins and Rhodamines. *Annu. Rev. Biochem* **2017**, *86*, 825-843.
11. Evans, J. C.; Nyquist, R. A., The Vibrational Spectra of Ethynyl Benzene and Ethynyl Benzene-D. *Spectrochim. Acta* **1960**, *16* (8), 918-928.

REVIEWERS' COMMENTS

Reviewer #1 (Remarks to the Author):

According to the comments from reviewers, the manuscript from Min et al has been properly revised. Although this reviewer has been supportive for this interesting manuscript which established robust methods for preparing a series of 9-cyanopyronins with various functionalities, I am now totally convinced that this manuscript is suitable for publication in Nature Communication. Especially, newly added data including RIE values and stability studies in aqueous solution should be informative and helpful for the readers. I also understand that maximum number for multiplexed immunostaining is limited at present by the number of commercially available secondary antibodies, although I expect that the authors would be able to perform it by using their newly devised "functionalized" Raman probes for cellular imaging in future. Anyway, I believe that the revised manuscript should be of interest to a broad readership for Nature Communications.

Reviewer #2 (Remarks to the Author):

The authors have addressed my concerns. I recommend that this excellent paper be published in Nature Communications.

Response to Reviewer 1:

According to the comments from reviewers, the manuscript from Min et al has been properly revised. Although this reviewer has been supportive for this interesting manuscript which established robust methods for preparing a series of 9-cyanopyronins with various functionalities, I am now totally convinced that this manuscript is suitable for publication in Nature Communication. Especially, newly added data including RIE values and stability studies in aqueous solution should be informative and helpful for the readers. I also understand that maximum number for multiplexed immunostaining is limited at present by the number of commercially available secondary antibodies, although I expect that the authors would be able to perform it by using their newly devised “functionalized” Raman probes for cellular imaging in future. Anyway, I believe that the revised manuscript should be of interest to a broad readership for Nature Communications.

We thank the reviewer for the understanding and very supportive comments.

Response to Reviewer 2:

The authors have addressed my concerns. I recommend that this excellent paper be published in Nature Communications.

We thank the reviewer for the very supportive comments.